# Radiotherapy and Systemic Therapies: Focus on Head and Neck Cancer

**DOI:** 10.3390/cancers15174232

**Published:** 2023-08-24

**Authors:** Francesca De Felice, Carlo Guglielmo Cattaneo, Pierfrancesco Franco

**Affiliations:** 1Radiation Oncology, Policlinico Umberto I, Department of Radiological, Oncological and Pathological Sciences, “Sapienza” University of Rome, 00161 Rome, Italy; carloguglielmo.cattaneo@uniroma1.it; 2Department of Translational Medicine (DIMET), University of Eastern Piedmont, Department of Radiation Oncology, “Maggiore della Carità” University Hospital, 28100 Novara, Italy

**Keywords:** head and neck cancer, radiation therapy, chemoradiotherapy, immunotherapy, p16, survival, response

## Abstract

**Simple Summary:**

This review provides some highlights in the current radio(chemo)therapy strategy for head and neck squamous cell carcinoma (HNSCC). The aim is to lead to a better understanding of this disease, potentially improving the standard of care and offering a starting point for reflection on future therapeutic developments.

**Abstract:**

Head and neck squamous cell carcinoma (HNSCC) is a complex clinical entity, and its treatment strategy remains a challenge. The best practice management for individual HNSCC patients should be discussed within a multidisciplinary team. In the locally advanced disease, radiation therapy (RT) with or without concomitant cisplatin-based chemotherapy is the current standard of care for most patients treated definitively or adjuvantly after surgery. Intensity-modulated photon therapy (IMRT) is the recommended RT technique due to its ability to offer considerable treatment conformality while sparing surrounding normal critical tissues. At present, the development of novel treatment strategies, as well as alternative systemic agent combinations, is an urgent need to improve the therapeutic ratio in HNSCC patients. Despite the immune landscape suggesting a strong rationale for the use of immunotherapy agents in HNSCC, evidence-based data demonstrate that combining RT with immune checkpoint inhibitors as the primary treatment modality has not been shown to induce significant benefit on survival clinical outcomes. The objective of this article is to review the current literature on the treatment of patients with HNSCC. We initially provided a comprehensive overview of the standard of care. We then focused on the integration of systemic therapies with RT, highlighting the latest published evidence and ongoing trials which investigate different combination strategies in the definitive setting. Our hope is to summarize relevant literature in order to provide a foundation for interpreting emerging data and designing future trials to maximize care, both in disease control and patient quality of life.

## 1. Introduction

Most head and neck squamous cell carcinomas (HNSCCs) arise from the epithelial cells of the upper aero-digestive tract, including the oral cavity, the oropharynx, the hypopharynx and the larynx. HNSCC has an estimated 744,994 new cancer cases (excluding nasopharynx, salivary glands and skin cancers) and 364,339 cancer deaths (excluding nasopharynx, salivary glands and skin cancers) worldwide [1]. In Europe HNSCC is the seventh most common type of cancer, with approximately 150,000 new patients diagnosed per year [2].

Since preventive *Make Sense campaign* is ultimately warranted [3], early detection of a potential malignant lesion in the head and neck region should be stressed. Chronic pain in the throat resistant to pharmacological therapy, persistent hoarseness and change in the voice, pain and/or difficulty in swallowing, the presence of non-healing mouth ulcers and/or red-white patches in the mouth, lumps in the neck and blocked nose on one side and/or bloody discharge from the nose are all symptoms that should lead physicians to refer a patient to a head and neck specialist. Once a malignant lesion is histologically proven, to define the most appropriate strategy—in order to obtain the highest cure rate and guarantee the best possible quality of life—it is necessary to take into consideration both the tumor’s (primary tumor location, TNM classification, HPV status) and the patient’s characteristics (age at diagnosis, comorbidities, family history of oncological pathologies, work, eating habits, lifestyle). Discussion within a multidisciplinary team should be prioritized [4]. The multidisciplinary team should include expert professionals characterized by long-time experience in care and/or clinical trials of HNSCC patients and/or scientific research [5].

Although most non-nasopharyngeal HNSCC cases (80%) are correlated to tobacco and alcohol (ab)use, human papillomavirus (HPV) infection is a growing HNSCC causal agent, particularly in oropharyngeal cancer [6]. Nowadays, HPV immunohistochemical evaluation is mandatory for all new diagnoses of squamous cell carcinomas of the oropharynx due to its prognostic value [6]. In fact, it has been demonstrated that patients with HPV-positive oropharyngeal cancer have a better survival rate than those with a similarly staged (according to the old classification, the 7th edition TNM staging system) HPV-negative disease [6]. However, the current TNM stratification system is not practice-changing. The use of novel prognostic/predictive biomarkers is urgently needed to enable better patient selection to better tailor the treatment in the future. For instance, a decision tree approach—based on the HPV status, the nodal involvement at diagnosis and the evidence of complete response within months from the end of treatment—has shown promise to identify patients who are at high (HPV-negative oropharyngeal cancer either with no evidence of complete tumor regression within 3 months and/or early responders but with cN ≥ 2c according to TNM 8th edition) or low (HPV-related cases) risk of death and thus are potential candidates for treatment intensification or deintensification, accordingly [7]. Treatment intensification in the setting of definitive/adjuvant radiation therapy (RT) can be achieved by dose escalation or “new” concomitant systemic treatment regimes. Incorporating recent improvements in diagnostics and RT techniques might facilitate treatment personalization, as well as ameliorate treatment-related toxicity profiles.

In this context, we hereby provide an overview of the HNSCC landscape. We performed a PubMed search using the keywords “radiation”, “radiotherapy”, “immunotherapy” and “head and neck cancer”. Only papers published in English were considered, and the discussed ones were manually chosen at the discretion of the authors. We focused on (i) the standard RT treatment, (ii) the role of induction chemotherapy and the update first-line treatment, (iii) the main changes and pitfalls with respect to the recent integration of systemic therapies with RT and (iv) the challenges of the next-generation clinical trials. These issues will be discussed with a summary of the current evidence base.

## 2. Radiation Therapy for Head and Neck Cancer

*Highlights.* Intensity-modulated RT (IMRT) plays a central role in the management of HNSCC patients and can be offered in a curative setting or as an adjuvant strategy or as part of a palliative treatment purpose [8]. IMRT is usually considered as definitive treatment in the oropharynx and larynx cancer sites, especially in locally advanced disease as an organ preservation strategy [8]. When appropriate, the addition of concomitant platinum-based chemotherapy (CRT) should be recommended due to its absolute 6.5% overall survival benefit [9]. In cases of HNSCC patients unfit for high-dose cisplatin (total dose of ≥200 mg/mq), hyperfractionated RT or RT plus concomitant cetuximab should be the standard of care [10,11].

The mainstay of treatment for oral cavity cancer is surgery followed by adjuvant RT—with or without concomitant chemotherapy—in the case of pathological T3-4, N2-3 nodal disease, positive surgical margins, extracapsular nodal spread, perineural and/or lymphovascular invasion [8]. 

Intensity-modulated proton therapy (IMPT) is associated with reduced rates of toxicity, but it is still not recommended as routine RT modality [8].

Standard treatment regimens are summarized in Table 1.

*Curative setting.* Early-stage disease—including T1-2N0 oral cavity, laryngeal, hypopharyngeal and p16-negative/positive oropharyngeal cancer [8]—can be treated by definitive RT alone. A conventional fractionation regimen (2 Gy per fraction) with a full therapeutic dose to the primary tumor of 70 Gy and a prophylactic dose to the nodal levels of 50 Gy is recommended. Although HPV-positive HNSCC is more radiosensitive and shows a favorable prognosis, the total dose for oropharyngeal cancer currently remains the same, regardless of HPV status [8].

Definitive CRT can be proposed in locally advanced HNSCC—including either stage III–IV oral cavity, larynx, hypopharynx and p16-negative oropharyngeal cancer or T3-4 N0-3, anyT N1-3 p16-positive oropharyngeal cancer [8]. Primary CRT is preferred for conservative strategy (in case of mutilating surgery) or in non-resectable lesions. Concomitant cisplatin-based CRT achieved higher locoregional control and improved overall survival compared to RT alone [9]. The largest benefit was observed with a total dose of cisplatin ≥200 mg/mq and in patients aged ≤ 70 years [9]. Two cycles of cisplatin (100 mg/mq) concurrently with accelerated RT (70 Gy in 6 weeks) showed similar survival outcomes with improved chemotherapy compliance compared to three cycles of high-dose cisplatin with a standard fractionation regimen (70 Gy in 7 weeks) [12]. Therefore, a total dose of ≥200 mg/mq cisplatin is recommended if primary combined concomitant CRT is indicated.

Low-dose once-a-week cisplatin (40 mg/mq) could represent an option in those patients considered unfit to tolerate high-dose cisplatin, despite the fact that direct comparisons between 3-weekly high-dose cisplatin versus weekly low-dose cisplatin are still lacking [13].

A regimen of fluorouracil plus carboplatin combined to conconcomitant RT can also be used, as its benefit was much the same as that which was noted for concomitant high-dose single-agent cisplatin in the MACH-NC meta-analysis (HR 0.78 vs. 0.84; *p*-value = 0.19) [9]. Carboplatin plus paclitaxel represents another combination chemotherapy regimen that can be proposed in the curative setting (assuring a clinical complete response rate of over 75% and a 3-year overall survival of 48%) [14].

RT with concomitant cetuximab (initial loading dose of 400 mg/mq and then weekly 250 mg/mq) represents an alternative to cisplatin-based CRT [11,15,16]. It improves locoregional disease control (47% vs. 34% at 3 years) and 5-year overall survival (45.6% vs. 36.4%) without increasing the rates of severe mucositis or dysphagia compared to RT alone [15]. Recently, two randomized phase III trials have directly compared survival outcomes for patients with p16-positive oropharyngeal HNSCC treated with either high-dose cisplatin-based CRT or concomitant cetuximab with RT [17,18]. Although these two trials enrolled slightly different patient populations—all-risk patients in the NRG Oncology RTOG 1016 trial, low-risk patients in the De-Escalate HPV trial—both trials demonstrated an overall survival benefit in favor of cisplatin-based CRT without difference in frequencies of severe toxicity (despite specific toxicities profiles differing between cisplatin and cetuximab regimen) [17,18]. 

These data confirmed the cisplatin-based CRT as the standard of care, as well as the importance of reaching a cumulative dose of cisplatin ≥200 mg/mq in locally advanced HNSCC, despite the fact that RT with concomitant cetuximab or concomitant cisplatin-based chemotherapy has never been compared in HPV-negative tumors. There is currently insufficient evidence regarding treatment de-escalation in patients with p16-positive oropharyngeal cancer. Omitting concomitant chemotherapy or replacing chemotherapy with cetuximab is not recommended [8]. Therefore, an adequate definition of unfit patient for cisplatin-based concomitant therapy is crucial. Cetuximab should be reserved for those patients considered unfit for cisplatin. At present, hyperfractionated RT alone represents another valid option for the treatment of locally advanced HNSCC. Compared to conventional fractionation RT, hyperfractionated RT was associated with a significant benefit on overall survival, with an absolute difference at 5 years of 8.1% (95% CI 3.4–12.8) and at 10 years of 3.9% (95% CI0.6–8.4) [10]. Similarly, a direct comparison between hyperfractionated RT and concomitant CRT remains to be specifically tested.

The use of proton therapy for head and neck cancer has improved significantly over the past decade, even in Europe, due to proton centers’ increased accessibility [19]. Concerning the head and neck region, at present, treatment recommendations are limited to HNSCC recurrence after previous RT, adenoid cystic carcinoma of salivary glands, sarcomas and cancer of paranasal sinuses [20]. Given its ability to reduce the integral dose to the patient and maintain highly conformal target coverage, proton therapy offers new opportunities for improving HNSCC care and research. Different clinical outcomes, including toxicity, survival, cost-effectiveness analysis as well as patient-reported outcomes are being tested prospectively in clinical trials. Adaptive planning methods and robust optimization of treatment fields have the potential to further improve normal tissue sparing [21].

This improved sparing of normal tissue may potentially enhance immunological response by preserving more viable circulating immune cells and more robust bone marrow regeneration, allowing more infiltration of antigen-presenting cells and effector T cells to initiate and amplify a durable tumor control response [22]. In this context, proton therapy represents a promising measure to reduce treatment-related toxicity and improve long-term quality of life, especially in long term HPV-positive oropharyngeal HNSCC survivors. This issue shows a strong interest in the radiation oncology community, but to date, no definitive conclusion can be drawn in the curative setting. 

Another attractive option is FLASH-RT, which involves the ultra-fast delivery of high-dose radiation (up to 100 Gy in one second) [23,24]. Several reports have been published showing the theoretical advantages of FLASH-RT over IMRT and proton therapy in head and neck cancer [23]. Such data are surely premature for clinical applicability, but FLASH-RT can hold great promises for HNSCC treatment in the near future.

*Adjuvant setting.* Surgery is usually preferred for oral cavity, paranasal sinus and T4 laryngeal cancers. Adjuvant RT is required to decrease the risk of locoregional relapse and should be started within 6–7 weeks after surgery [8]. It should be recommended in case of high-risk factors, including pathological T3-4, N2-3 nodal disease, positive/close surgical margins, extracapsular nodal spread, perineural invasion and lymphovascular invasion [8]. Based on risk factors, a total dose of 60–66 Gy to the tumor bed and 50 Gy to the surgical bed and subclinical disease is indicated using conventional fractionation. When indicated, the standard treatment is CRT with high-dose cisplatin (100 mg/mq). In the adjuvant setting, based on the results of the JCOG1008 trial [25], CRT with weekly cisplatin at a dose of 40 mg/mq can be used as a possible alternative in post-operative high-risk locally advanced HNSCC cases (pathological stages III, IVa or IVb—according to 7th TNM edition—microscopically positive margin and/or extranodular extension). In fact, the JCOG1008 phase II/III trial—which randomized patients with postoperative high-risk HNSCC cancer to receive either CRT with 3-weekly cisplatin (100 mg/mq) or with weekly cisplatin (40 mg/mq)—demonstrated that CRT with weekly cisplatin was non-inferior to 3-weekly cisplatin in terms of overall survival (HR 0.69, 99.1% CI, 0.374–1.273), with a favorable toxicity profile [25]. However, its statistical design, mainly linked to the relatively wide non-inferiority margin of 10% for 5-year OS (corresponding to a non-inferiority margin of HR of 1.32), limits the robustness of the results.

*Palliative setting.* Palliative RT should be considered in advanced HNSCC when curative treatment is not feasible, in those HNSCC cases medically unsuitable for standard RT or those patients who have widely metastatic disease [26]. The aim is to relieve/prevent locoregional symptoms, avoiding severe RT-related toxicity. There are no practice recommendations for appropriate palliative RT regimens in this setting. The use of hypofractionated should be preferred. A total dose of 30 Gy in 10 fractions and 20 Gy in 5 fractions could provide guidance for optimal fractionation choices in extensive painful primary lesion and bleeding cases, respectively, with the aim of minimizing treatment burden while achieving symptomatic relief.

## 3. Systemic Therapy for Head and Neck Cancer

*Highlights.* Induction chemotherapy with a taxane/platinum/5-fluoruracil regimen (three courses) is recommended as a laryngeal preservation strategy [8].

Currently, in Europe, the combination of chemotherapy (cisplatin or carboplatin plus 5-fluoruracil) plus pembrolizumab and pembrolizumab monotherapy is the standard-of-care first-line therapy for recurrent/metastatic HNSCC with a combined positive score (CPS) ≥ 1 [8,27]. EXTREME regimen (cisplatin or carboplatin plus 5-fluoruracil plus cetuximab) remains the standard of care for patients with HNSCC not expressing PD-L1 and patients with contraindications to anti-programmed death-1 (PD-1) inhibitors [8].

*Induction chemotherapy.* Induction chemotherapy with a taxane/platinum/5-fluoruracil (TPF) schedule should be considered only for fit patients with high risk of distant metastases occurrence because it may be associated with a high risk of treatment-related death due to an inappropriate selection of patients or use of granulocyte-colony stimulating factor (G-CSF) [8]. The addition of induction chemotherapy in the management of non-laryngeal HNSCC can be associated with a substantial increase in CRT acute and late toxic effects, without a significant benefit for locoregional control and overall survival. The major criticism is that the vast majority of trials that have directly compared TPF induction chemotherapy followed by CRT with concomitant CRT included patients treated with TPF, followed by non-standard concurrent CRT regimens (such as weekly docetaxel at 20 mg/m^2^ for 4 weeks or carboplatin AUC 1,5 for 7 weeks) [28,29]. Therefore, the true value of induction chemotherapy has remained debatable over the years.

In this context, the second update of the Meta-Analysis of Chemotherapy in Head and Neck Cancer (MACH-NC) provided a robust analysis of the role of induction chemotherapy in non-nasopharyngeal HNSCC management [30]. The effect of induction chemotherapy was evaluated in 45 trials (7054 patients) with a median follow-up of 5.7 years. Induction chemotherapy followed by (C)RT, irrespective of tumor response, for non-laryngeal/non-hypopharyngeal tumors has not been shown to be superior to concomitant CRT alone. The overall survival benefit was 0.96 (95% CI 0.90–1.01; *p* = 0.14) with an absolute benefit of 2.2% at 5 years and 1.3% at 10 years compared to CRT alone [30]. Concomitant versus induction chemotherapy comparisons included 1214 patients. All survival endpoints demonstrated results in favor of CRT, with an absolute overall survival benefit of 6.2% at 5 years, an absolute event-free survival benefit of 3.7% at 5 years and an absolute locoregional failure benefit of 5.8% at 5 years [30].

*First-line therapy systemic therapy.* The development of immunotherapy, primarily based on check-point inhibitors, has revolutionized the management of different human malignancies, including recurrent/metastatic non-nasopharyngeal HNSCC [8,29,31]. The landmark KEYNOTE-048 clinical trial established pembrolizumab (a PD-1 inhibitor) with and without chemotherapy as the new standard first-line treatment for patients with platinum-sensitive recurrent/metastatic non-nasopharyngeal HNSCC with CPS ≥ 1 [29]. In this randomized phase III study, 882 patients with untreated locally incurable recurrent/metastatic non-nasopharyngeal HNSCC were randomized to receive pembrolizumab alone, pembrolizumab with chemotherapy or cetuximab with chemotherapy [29]. The study was powered to compare (i) pembrolizumab monotherapy with standard of care (cetuximab with chemotherapy) and (ii) pembrolizumab plus chemotherapy with standard of care (cetuximab with chemotherapy). Pembrolizumab with chemotherapy improved overall survival versus cetuximab with chemotherapy in the total population (13.0 months vs. 10.7 months, *p* = 0.0034) at the second interim analysis and in the CPS ≥ 20 patients (14.7 vs. 11.0, *p* = 0.0004) and CPS ≥ 1 patients (13.6 vs. 10.4, *p* < 0.0001) at final analysis [29]. Pembrolizumab alone improved overall survival versus cetuximab with chemotherapy in the CPS ≥ 20 patients (median 14.9 months vs. 10.7 months, *p* = 0.0007) and CPS ≥ 1 patients (12.3 vs. 10.3, *p* = 0.0086), and it was non-inferior in the total population (11.6 vs. 10.7) at the second interim analysis [29]. Neither pembrolizumab alone nor pembrolizumab with chemotherapy improved progression-free survival or objective response compared with cetuximab with chemotherapy [29]. At the final analysis, grade ≥ 3 adverse events occurred in 55% of the pembrolizumab alone group, 85% of the pembrolizumab with chemotherapy group and 83% of the cetuximab with chemotherapy group [29]. 

With regard to this and considering the differences in the toxicity profile of pembrolizumab alone versus in combination with chemotherapy, prioritizing discussion in a multidisciplinary team as well as patient preference should be prioritized in this treatment setting (recurrent/metastatic non-nasopharyngeal HNSCC expressing PD-L1), whereas the EXTREME regimen, including platinum, 5-fluoruracil and cetuximab, is still recommended for patients with recurrent/metastatic HNSCC not expressing PD-L1.

Based on the CheckMate 141 trial, nivolumab (a PD-1 inhibitor) is recommended in subsequent line for patients with recurrent/metastatic non-nasopharyngeal HNSCC with ongoing disease progression or after platinum-containing therapy (within 6 months) in the adjuvant or primary (with radiation) setting [29,32]. 

Patients were randomly assigned (2:1) to receive nivolumab or a standard, single-agent investigator’s choice of chemotherapy. Overall survival was significantly longer with nivolumab therapy than with standard therapy (HR 0.70, 97.73% CI 0.51–0.96; *p* = 0.01) [33]. The long-term 2-year analysis of CheckMate 141 reinforced nivolumab as the first-line treatment for patients with platinum-refractory recurrent/metastatic non-nasopharyngeal HNSCC [33]. 

## 4. Integration of Systemic Therapies with Radiotherapy

*Highlights.* Integration of systemic therapies with RT is a way to improve the therapeutic index ratio. The effect of targeted therapies is far from desirable, probably due to the HNSCC genetic heterogeneity. Reinforcing the standard of care with immunotherapy in the curative setting produced conflicting results. Further research is needed to improve the care of HNSCC patients.

*Target therapy.* Despite multimodal approaches and advances in the staging of patients, toxicity management strategies and radiation techniques, there is still a lack of significant improvement in HNSCC patients’ survival. Over the last decades, a great success has been achieved in targeted therapy, focusing on agents targeting different signaling pathways, such as epidermal growth factor receptor (EGFR) and phosphatidylinositol 3-kinase (PI3K) [11,34].

Concerning integration with RT for the treatment of locally advanced HNSCC, the association of RT plus EGFR inhibitors did not confer any survival benefit and seemed to increase toxicity compared to the standard cisplatin-based CRT [11,35]. Cetuximab can be the choice radiosensitizer in those patients not eligible for CRT [11]. Similarly, panitumumab cannot replace cisplatin in the combined treatment with RT [35]. 

*Immunotherapy.* Relevant prospective trials testing combined immunotherapy and RT in curative-intent treatment are summarized in Table 2 [36,37,38,39,40,41,42,43,44,45]. To note, to provide an overview of clinical trials, ongoing clinical trials were screened on “clinicaltrials.gov”. Overall, these trials mainly address (i) treatment escalation in intermediate-high-risk patients and (ii) treatment de-escalation in moderate-low-risk patients. Despite the efficacy observed in the recurrent/metastatic setting, the role of immune checkpoint inhibitors in the curative HNSCC setting remains to be determined, as well as the optimal timing, either concurrent or sequential. Several recent randomized clinical trials failed to demonstrate the advantage of adding concurrent PD-1 inhibitor to CRT in locally advanced disease [36,37,38]. Plausible explanations include the larger-volume elective nodal irradiation that might hinder immunotherapy therapeutic effects by directly depleting T cells. Future clinical trials of immunochemoradiotherapy may consider reducing the elective nodal volumes, even if this hypothesis must be carefully evaluated to minimize the risk of compromising decrelocoregional control.

The timing/sequencing between immunotherapy and CRT may also significantly impact response to exert synergistic efficacy. Allowing for time-recovery of the immune response prior to immune checkpoint inhibitors could result in a more favorable clinical outcome. Of note, a recent phase II trial showed a numerically superior 1-year and 2-year progression-free survival in sequential pembrolizumab (started two weeks after CRT) compared to concurrent pembrolizumab (started one week prior to CRT) [46]. This clinical activity will be tested in a phase III trial.

Newer radiosensitizers have been evaluated in locally advanced non-nasopharyngeal HNSCC. The Debio 1143-201 phase II trial demonstrated the efficacy of xevinapant (debio 1143) in association with standard cisplatin-based CRT [47]. Patients were randomly assigned to receive xevinapant plus standard CRT or placebo plus standard CRT. Xevinapant is a pro-apoptotic agent that inhibits the inhibitor of apoptosis proteins. Locoregional control at 18 months was achieved in 54% of patients of the xevinapant group versus 33% of the CRT alone group (*p* = 0.026), without affecting treatment compliance or compromising patient safety [47]. A confirmatory phase III study is ongoing.

## 5. Conclusions and Future Directions

In patients with locally advanced HNSCC, cisplatin-based CRT is a critical component of curative therapy. There remains a significant clinical gap in improving its efficacy, and the development of non-overlapping strategies to improve treatment outcomes is needed. The main question remains whether survival outcomes can be reliably improved by the combination of standard CRT and immunotherapy in the curative/adjuvant setting. With the increasing RT technical power to safely increase local HNSCC control, it is the right time to carefully explore the potential benefits of combining RT with new molecular targeting and immunotherapy agents to improve HNSCC survival outcomes. While the advent of immune checkpoint inhibitors offers a robust opportunity to improve the first-line strategy for recurrent/metastatic non-nasopharyngeal HNSCC, combining pembrolizumab or nivolumab with radical RT cannot be used to replace a cisplatin-based agent as the standard of care in combined treatments. Adding novel immunotherapies such as a pro-apoptotic agent (xevinapant) in combination with RT may yield improved results beyond what is currently achievable.

Certainly, the timing and sequencing of RT and immunotherapy remains a subject of debate. RT can substantially alter tumor microenvironment and trigger mechanisms of resistance [48,49,50,51]. Therefore, understanding the temporal framework for immune response after RT fractions is crucial for optimal sequencing of RT and immunotherapy. Much should be learned from preclinical orthotopic models that recapitulate human HNSCC disease. In addition, at present, there are no biomarkers to guide non-nasopharyngeal HNSCC treatment in clinical practice. A more comprehensive understanding of non-nasopharyngeal HNSCC biology is needed. Biomarker-driven clinical trials should be realized, and platform trials should be promoted. Newer radiosensitizers, technologies and predictive signatures, such as microbiome [49], could promote individualized RT. By taking advantage of *omics* sciences, treatment personalization for non-nasopharyngeal HNSCC patients should be a shared goal to improve outcomes while reducing complications. This overview may help to guide clinical decision making and assist researchers in the design of future clinical and translational trials. 

## Figures and Tables

**Table 1 cancers-15-04232-t001:** Standard treatment regimens for HNSCC.

Setting	Regimen		Remarks
	Radiotherapy	Concomitant Chemotherapy	
Curative	conventional	cisplatin 3-weekly 100 mg/m^2^	preferred regimen
	conventional	cisplatin weekly 40 mg/m^2^	Has not been compared head-to-head with cisplatin 100 mg/m^2^
	conventional	cetuximab (loading dose 400 mg/m^2^; then weekly 250 mg/m^2^)	patients unfit for platinum
	hyperfractionated	-	has not been compared head-to-head with CRT
Adjuvant	conventional	cisplatin 3-weekly 100 mg/m^2^	preferred regimen
	conventional	cisplatin weekly 40 mg/m^2^	noninferior to cisplatin 3-weekly
Palliative	hypofractionated	-	no specific recommendations

CRT: chemoradiotherapy.

**Table 2 cancers-15-04232-t002:** Relevant prospective trials testing combined immunotherapy and RT in curative-intent treatment.

Trial Identifier	Trial Name	Phase	Patient Population	N Planned	Recruitment Status	Treatment	Primary Outcome
*Treatment escalation*							
NCT 02952586 [36]	JAVELIN HN 100	III	SCC of oral cavity, oropharynx, hypopharynx, or larynx (HPV negative disease, Stage III, IVa, IVb; non-oropharyngeal HPV positive disease Stage III, IVa, IVb, HPV positive oropharyngeal disease T4 or N2c or N3)	697	Terminated, has results	Avelumab + SoC CRT	PFS
						Placebo + SoC CRT	Not met (HR 1.21, 95% CI 0.93–1.57)
NCT 03040999 [37]	KEYNOTE-412	III	SCC of larynx, hypopharynx, p16-negative oropharynx, or oral cavity T3–T4 (N0–N3) or T1–T4 (any N2a-3); p16-positive oropharyngeal cancer T4 or N3	804	Active, not recruiting	Pembrolizumab + CRT (either accelerated or standard fractionation)	EFS
						Placebo + CRT (either accelerated or standard fractionation)	
NCT 02707588 [38]	GORTEC 2015-01 PembroRad	II	SCC of oral cavity, oropharynx, hypopharynx and larynx (Stage III, IVa and Ivb)	133	Active, not recruiting	Pembrolizumab and RT	LRC
						Cetuximab and RT	
NCT 03258554 [39]	NRG-HN004	II/III	SCC p16-positive oropharyngeal, unknown primaries stage III and selected stage I–II based on smoking status in pack-years; SCC laryngeal, hypopharyngeal, oral cavity, p16-negative oropharyngeal, unknown primaries stage III–IVB	493	Active, not recruiting	Cetuximab and RT	PFS (phase II), OS (phase III)
						Durvalumab and RT	
NCT 02999087 [40]	REACH	III	SCC of oral cavity, oropharynx, hypopharynx or larynx stage III, stage IVa (operable, but not operated) or IVb (non resectable)	688	Active, not recruiting	SoC CRT (fit patients)	PFS
						Avelumab, cetuximab and RT (fit patients)	
						SoC CRT (unfit patients)	
						Avelumab, cetuximab and RT (unfit patients)	
NCT 03811015 [41]	EA3161	III	SCC p16-positive oropharyngeal ≥ 10 pack-years, stage T1-2N2-N3 or T3-4N0-3; or < 10 pack-years, stage T4N0-N3 or T1-3N2-3	636	Recruiting	SoC CRT and adjuvnt nivolumab	OS
						SoC CRT and osbervation	
						Nivolumab (if patients progress within 12 months from CRT)	
NCT 03576417 [42]	NIVOPOSTOP	III	SCC of oral cavity, oropharynx, hypopharynx or larynx pStage III–IV (oropharyngeal cancer pStage II p16 positive with pT3N1 or pT4N1 and tobacco consumption ≥ 20 packs/year)	680	Recruiting	SoC CRT and osbervation	DFS
						SoC CRT and nivolumab for maintenance	
NCT 01810913 [43]	RTOG 1216	II/III	SCC of oral cavity, oropharynx (p16 negative), larynx, or hypopharynx pStage III-IV	613	Recruiting	SoC CRT	DFS (phase II), OS (phase III)
						Docetaxel and RT	
						Docetaxel, cetuximab and RT	
						Atezolizumab and SoC CRT	
NCT 03452137 [44]	IMvoke010	III	SCC of the head and neck, completed definitive local therapy	406	Active, not recruiting	Adjuvant atezolizumab	EFS
						Adjuvant placebo	
*Treatment de-escalation*							
NCT 03952585 [45]	NRG-HN005	II/III	SCC of oropharynx T1-2, N1 or T3, N0-N1	711	Recruiting	CRT (accelerated fractionation)	PFS
						CRT (standard fractionation)	
						Nivolumab + CRT (accelerated fractionation over 6 fractions/week for 5 weeks)	

SCC: squamous cell carcinoma; SoC: standard of care; CRT: chemoradiotherapy; PFS: progression-free survival; EFS: event-free survival; LRC: locoregional control; DFS: disease-free survival.

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
