# Peer review of "Radiotherapy and Systemic Therapies: Focus on Head and Neck Cancer"

_cancers, 2023, doi:10.3390/cancers15174232_

Round 1
Reviewer 1 Report
Thank you very much for the very well written review.
I have the following remarks:
Non-nasopharyngeal cancers terminology could be helpful
For the curative setting, alternative chemo prtocols were overlooked although their effectiveness. At least Carbo+5FU. Furthermore, the fractionated cisplatin protocol (100mg W1 and W5 over 5 days). Both protocols have more solid results other than the weekly cisplatin. Another Protocol which could be comparible to weekly cis. is the carbo-pacli.
After induction Chemotherapy, the concurrent chemotherapy limitations and options are worth discussion.
Author Response
Thank you very much for the very well written review.
I have the following remarks:
- Non-nasopharyngeal cancers terminology could be helpful
• Text has been modified accordingly.
- For the curative setting, alternative chemo prtocols were overlooked although their effectiveness. At least Carbo+5FU. Furthermore, the fractionated cisplatin protocol (100mg W1 and W5 over 5 days). Both protocols have more solid results other than the weekly cisplatin. Another Protocol which could be comparible to weekly cis. is the carbo-pacli.
• Text has been modified accordingly.
- After induction Chemotherapy, the concurrent chemotherapy limitations and options are worth discussion.
• Text has been modified accordingly.
- Abstract is generally one paragraph, it is unusual to divide the abstract into two parts.
• We agree. We followed the “journal style” (a simple summary should be written in one paragraph before the Abstract). But if it not mandatory we can remove it.
Reviewer 2 Report
This (quite) short review is very useful for those who treat head and neck cancers and even for those oncologists who just want ti refresh knowledge on this specific field. This article also highlight p16 positive tumors usually affected people living in heterogen situation... I am happy to read the weekly cisplatin regimen that is non-inferior to 3weekly high dose chemotherapy, this is more human therefore I prefer and hope worldwide this would be spread. I highly recommend to publish this article.
Author Response
Thank you